# Field Testing Two Flux Footprint Models

Trevor W. Coates[1], Monzurul Alam[2], Thomas K. Flesch[3], Guillermo Hernandez-Ramirez[2]

[1] Agriculture and Agri-Food Canada, Lethbridge, Canada, T1J 4B1
[2] Department of Renewable Resources, University of Alberta, Edmonton, Canada T6G 2E3
[3] Department of Earth and Atmospheric Sciences, University of Alberta, Edmonton, Canada T6G 2E3
*Correspondence to*: Thomas Flesch (thomas.flesch@ualberta.ca)

**Abstract**

A field study was undertaken to investigate the accuracy of two micrometeorological flux footprint models for calculating the gas emission rate from a synthetic 10 x 10 m surface area source, based on the vertical flux of gas measured at fetches of 15 to 50 m downwind of the source. Calculations were made with an easy to use tool based on the Kormann-Meixner analytical model and with a more sophisticated Lagrangian stochastic dispersion model. A total of 59 testable 10 minute observation periods were measured over nine days. On average, both models underestimated the actual release rate by approximately 30%, mostly due to large underestimates at the larger fetches. The accuracy of the model calculations had large period-to-period variability, and no statistical differences were observed between the two models in terms of overall accuracy.

**1 Introduction**

Micrometeorological techniques such as eddy covariance and flux-gradient measure a vertical flux of gas in the atmosphere, which can be used to deduce the flux from an underlying surface area of interest. If the underlying surface is expansive and horizontally homogenous, the measured atmospheric flux and the surface flux can be considered equivalent (Dyer, 1963). However, if the area of interest has a limited spatial extent, or is located some distance from the atmospheric measurement, the relationship between the two fluxes can be complex, as the measured flux may be capturing a dynamic mixture of surface fluxes from both inside and outside the area of interest. In these cases, flux footprint modelling can be used to quantify the relationship between the measured atmospheric flux and the surface flux from the area of interest.

The analytical flux footprint model of Kormann and Meixner (2001), hereafter referred to as the KM model, is widely used to evaluate and interpret flux measurements taken over spatially limited surface sources. The KM model relies on a simplified representation of atmospheric transport (Schmid, 2002) to create an easily computable footprint. It has been used to help quantify ammonia fluxes from fertilized plots (Spirig et al., 2010), interpret methane fluxes from heterogeneous peatland areas (Budishchev et al., 2014), and to reject periods where the footprint extends outside the source of interest (Stevens et al., 2012). Other footprint models use a more realistic treatment of atmospheric transport (e.g., Kljun et al., 2002; Sogachev and Lloyd, 2004). Using a state of the art Lagrangian stochastic (LS) footprint model, Wilson (2015) found a clear separation between the footprints computed with the LS and KM models, depending on atmospheric stability and the distance from the measurement location. While more rigorous footprint models are clearly more defensible, the simpler KM model has the advantage of rapid analysis and the existence of software tools that make its application more accessible to non-specialists (Neftel et al., 2008).

This field study compares the accuracy of the KM footprint model with a more rigorous LS model. The motivation for this study was the question of whether the accuracy of the LS model was sufficiently better than the KM model so as to justify a more complex LS application. In this experiment we released gas at a known rate from a small synthetic area source and measured the vertical gas flux at a downwind location using the eddy covariance technique. The KM and LS models were then used to calculate the source emission rate from the measured atmospheric flux. The accuracy of those calculations is examined in this report. This follows the approach of Heidbach et al. (2017) and Coates et al. (2017) in their experimental evaluation of footprint models.

## 2 Methods

### 2.1 Gas Release

The experiment took place on an extensive, flat agricultural field at the University of Alberta's Breton Research Farm, in Alberta, Canada (53° 07′ N, 114° 28′ W). Measurements were made after autumn harvest, and the surface was rye (*Secale cereale* L.) stubble with an average height of 3 cm. No obstructions to the wind were present within 250 m of the measurement site.

A synthetic source of carbon dioxide ($CO_2$) gas was constructed using 10 lengths of ½" (12.7 mm) diameter PVC pipe, each 10 m long. The 10 pipes were loosely positioned to create a nominal 10 x 10 m square source area. Compressed $CO_2$ gas (99.9 % purity) passed through a mass flow controller (GFC57 configured for $CO_2$, Aalborg Instruments and Controls, Inc. Orangeburg, NY, USA) to a manifold (17 L) having outlets for each of the 10 pipes. Gas outlets of 1/64" (0.4 mm) diameter were placed every 50 cm along each pipe. We assumed equal flow rates from each outlet, which requires the gas outlets be identical and the pressure loss across each outlet to be much greater than the pressure loss along the source piping (Flesch et al., 2004). We estimated pressure losses using simplified equations for pipe flow, assuming incompressibility and a re-entrant type outlet shape (Fox and McDonald, 1985). For our most commonly used release rate of 90 L min$^{-1}$, the pressure loss across the outlets is approximately 5,000 Pa whereas the loss along a 10 m pipe section is only approximately 40 Pa.

The vertical $CO_2$ flux downwind of the synthetic source was measured using the eddy covariance (EC) technique. The instrumentation included a fast-response $CO_2$/$H_2O$ analyser (Li-7500DS, Licor Biosciences, Lincoln, NE, USA) and a sonic anemometer (CSAT-3, Campbell Scientific, Logan, UT, USA) co-located at a height of 1.97 m above ground. The 10 Hz concentration and wind measurements were processed using the EddyPro® open source software (version 6.2.1 LI-COR Biosciences, Lincoln, NE, USA) to obtain 10 minute (min) average fluxes of $CO_2$. The flux calculation applied a double coordinate wind rotation, Webb-Pearman-Leuning correction terms for density fluctuations (Webb et al., 1980), and spectral corrections for inadequate high and low frequency response of the sensors (Moncrieff et al., 1997, 2004). Quality checks for steady state conditions and integral turbulence characteristics were used to exclude error-prone periods (Foken and Wichura, 1996).

Gas releases took place over nine days, with the center of the synthetic source positioned (Fig. 1) at one of three nominal distances from the EC system (fetches of 15, 30, and 50 m). Placement of the source relative to the EC system depended on the expected wind direction. Because $CO_2$ is naturally emitted from the landscape it was important that the synthetic $CO_2$ release rate be sufficiently high so as to create a measured atmospheric flux that was many times larger than the natural landscape flux. Nicolini et al. (2017) found a $CO_2$ release rate of 22 L min$^{-1}$ was sufficient to distinguish the release signal from background levels. Our situation was helped in that the experiment took place during the dormant autumn season when landscape $CO_2$ fluxes were small. Gas was released at rates between 30 and 90 L min$^{-1}$, with larger rates used for the larger fetches. Prior to any release interval, and immediately after each hour of gas release, a 30 min period of background $CO_2$ flux was measured. These background fluxes (which were consistently small) were subtracted from the EC measured fluxes prior to undertaking the footprint analyses.

Our study consisted of more than 300 10 min flux measurement periods, and included periods of gas release, background flux measurements, and transitions when gas was released but a steady state plume may not have been established over the field site (we assumed this occurred 10 min after gas was turned on). There was a total of 125 valid gas release periods. From this total we excluded 66 periods from our analysis based on two broad factors:

- 19 periods were excluded for having wind conditions associated with unreliability in the EC measurements or the dispersion model calculations: light winds with a friction velocity $u* < 0.05$ m s$^{-1}$, or an inferred roughness length $z_0 > 0.25$ m. A low $u*$ filtering criterion is often used in EC analyses (e.g., Rannik et al., 2004) and in dispersion model

calculations (e.g., Flesch et al., 2014). The $z_0$ filtering criterion indicates an unrealistic wind profile given the bare soil

conditions of our site, and the likelihood of inaccurate dispersion model calculations given that wind profile.

•   47 periods were excluded when the EC measurement location was not obviously in the source plume. This included

periods when the measured $CO_2$ flux was less than zero, when the wind direction deviated more than 30 degrees from the

line between the EC site and the source center, or when the LS footprint model (described below) indicated the plume

may not have reached the EC measurement site (i.e., fewer than 1,000 of 1,000,000 backward trajectories released from

the EC site reached the source).

These quality control criteria eliminated over half of the gas release periods, leaving 59 periods for the footprint analysis. The
final data are provided in the supplemental material accompanying this report.

**2.2 Flux Footprint Models**

**2.2.1 Kormann and Meixner (KM) Model**

The KM model is based on an analytical solution to the steady-state advection-diffusion equation, assuming simplified power-law
profiles for windspeed and eddy diffusivity, and a crosswind diffusion component (Kormann and Meixner, 2001). We used the
ART Footprint Tool software (Spirig et al., 2007) based on the KM model to calculate the synthetic source emission rate ($Q_{KM}$, g
C m$^{-2}$ s$^{-1}$) from the measured EC flux. The calculation uses the spatial outline of the source polygon, the EC measurement height
($z_{EC}$), the horizontal wind speed at height $z_{EC}$, the friction velocity ($u_*$), the standard deviation of the lateral wind velocity ($\sigma_v$), and
the Obukhov length ($L$). The wind variables were measured with a 3-D sonic anemometer (part of the EC system). In this study,
the ratio of the KM-calculated synthetic emission rate to the actual release rate ($Q_{KM}/Q$) is the metric for model testing. A perfectly
accurate calculation gives $Q_{KM}/Q = 1$.

**2.2.2 Lagrangian Stochastic (LS) Model**

A state of the art LS model was also used to calculate the emission rate from the synthetic source ($Q_{LS}$, g C m$^{-2}$ s$^{-1}$) based on the
measured EC flux. The relationship between the source emission rate and the EC flux was calculated from the trajectories of
thousands of model "particles" travelling upwind from the EC measurement point (backward in time). We follow the calculation
procedure outlined in Flesch (1996) using the LS model detailed in Flesch et al. (2004). This model uses the wind velocity
fluctuations in the three directional components ($\sigma_u$, $\sigma_v$, $\sigma_w$), the friction velocity ($u_*$), the Obukhov stability length ($L$), the average
wind direction, and the surface roughness length ($z_0$). These properties were calculated from the 3-D sonic anemometer
measurements. The LS calculations were made using 1,000,000 particles for each 10 min observation interval. A perfectly accurate
LS model calculation gives $Q_{LS}/Q = 1$.

**2.3 Statistical Analysis**

The accuracies of the footprint calculations are evaluated from the ratio of the model calculated emission rate to the actual release
rate: $Q_{KM}/Q$ and $Q_{LS}/Q$. These ratio data are asymmetrically distributed, and a logarithmic transform of the ratios is used when
making our statistical comparisons. Thus, the geometric means of the emission ratios is our measure of central tendency.
Confidence intervals for the geometric mean are calculated using the log-transformed ratio data, and then converted back to ratio
units (Limpert et al., 2001). The confidence intervals (CI) are asymmetrical, and we report the upper and lower limits of the
intervals.

## 3 Results and Discussion

The synthetic emission rates calculated with both footprint models underestimate the actual emissions by roughly 30% on average. The overall means of the footprint calculations, expressed as the ratio of the model calculated emission rate to the actual emission rate, are $Q_{KM} / Q = 0.67$ (95% CI: 0.50, 0.89) and $Q_{LS} / Q = 0.77$ (CI: 0.60, 0.98). These means are statistically less than 1.0, but not different from each other (paired t-tests with $P$s > 0.05). The period-to-period variability in the $Q / Q$ ratios is large, with $Q_{KM} / Q$ ranging between 0.04 and 2.20 and $Q_{LS} / Q$ between 0.06 and 4.44. Some of the variability is likely due to the small size of the area source. The 10 x10 m source covers a small portion of the entire flux footprint. As opposed to larger source areas, the small area should amplify the differences between the models, and increase the relative uncertainty in the footprint calculations (i.e., increasing the size of the source area means increasing the spatial integration of the footprint function in the calculations, which acts to increasingly constrain the $Q / Q$ values closer to one).

When examining the footprint agreements as a function of fetch (Fig. 2), we find both models are accurate at the shorter fetch of 15 m, as the means of $Q_{KM} / Q$ and $Q_{LS} / Q$ are not statistically different from 1. At the 15 m fetch the $Q_{KM}$ calculation tends to slightly overestimate the actual emission rate with $Q_{KM} / Q = 1.17$ (CI: 1.00, 1.36), while $Q_{LS}$ tends to slightly underestimates it with $Q_{LS} / Q = 0.84$ (CI: 0.68, 1.04). Based on the calculations of Wilson (2015) and Heidbach et al. (2017), we had hypothesized that there would be substantial differences between the two models at the shorter fetch, with the LS model being more accurate than KM due to a better representation of horizontal turbulent transport, which is particularly important for defining the footprint at short fetches. However, this is not the case in this study. At the intermediate fetch of 30 m, the KM model slightly overestimates the emission rate with $Q_{KM} / Q = 1.21$ (CI: 0.86, 1.71), while the LS model substantially overestimates it with $Q_{LS} / Q = 1.75$ (CI: 1.39, 2.21). At the larger 50 m fetch, both models substantially underestimate the emission rate, with $Q_{KM} / Q = 0.29$ (CI: 0.17, 0.51) and $Q_{LS} / Q = 0.51$ (CI: 0.30, 0.86). The underestimate of $Q_{KM} / Q$ at the larger fetch is similar to findings by Tallec et al. (2012) and Felber et al. (2015). At the 50 m fetch the measured EC fluxes were smaller than was measured at the shorter fetches, and in some cases the measured flux fell to a level near the background landscape flux (e.g., five periods had a measured flux that was less than five times the magnitude of the background flux). This was despite maximizing the gas release rate for the larger fetches. The result is that for the larger fetches there is increased measurement uncertainty (relative) in the flux signal from the gas release, and increased uncertainty in $Q_{KM}$ and $Q_{LS}$. Some of the relative uncertainty we see in $Q / Q$ for the 50 m fetch is likely due to this factor.

In Figure 3 we show the $Q / Q$ ratios grouped according to atmospheric stability. The observations are separated into three groups having nearly equal numbers of observations: neutral ($|L| > 60$ m), unstable ($0 > L > $ -60 m), and stable ($60 > L > 0$ m). For the neutral and unstable groups, the mean $Q / Q$ from both models does not statistically differ from 1, nor does it differ between groups due to the large variability in the calculations. However, in stable conditions both models are inaccurate and they substantially underestimate the actual emission rate. A more detailed look at the stable cases shows the $Q_{KM} / Q$ calculations are particularly inaccurate for the 50 m fetch, with a mean of 0.14 (CI: 0.03, 0.62).

There are no clear patterns in terms of explaining the differences between the two footprint models based on environmental factors. Whether we separate the data by fetch or by stability, the results from the two models are not statistically different from each other. Wind speed, roughness length, and wind direction (deviation from a line between the EC system and the source) were also considered as factors to explain the model differences, but again, no pattern was observed. The lack of model differences was unexpected given the studies of Göckede et al. (2005), Wilson (2015), and Heidbach et al. (2017) showing large differences in the calculations between analytical and LS models. This suggests that in our study, any systematic differences between the models

were obscured by the substantial period-to-period variability in the $Q/Q$ calculations, and that the detection of model differences
would require a much larger observational sample size than we were able to acquire.

## 4 Conclusions

From an end-user's perspective, our results show that both the KM and LS models returned reasonably accurate flux footprint
estimates on average, particularly for the shorter measurement fetches. Our dataset does not consistently discriminate between the
performance of the two models, despite the theoretical advantages of the LS model. Based on the results of this study, we conclude
that the easy to use KM model can provide accurate footprint calculations that are accessible to non-specialists.
It is clear that the KM and LS footprint models give systematically different results (as shown in Wilson 2015); but that we were
unable to (statistically) observe these differences given the large period-to-period variability in the calculations and the relatively
small number of field observations. The small area of our synthetic source likely contributed to the large variability, and a larger
source may have allowed better differentiation between the models. However, period-to-period variability is the nature of footprint
calculations based on simplified models of atmospheric transport like the KM and LS formulations. These model calculations,
which at best approximate an ensemble average realization of the atmosphere, will not reflect the period-to-period fluctuations of
actual measurement periods.

## Data availability

The data used in this analysis are available as supplementary material, or by request to Trevor.Coates@Canada.ca.

## Author contributions

TC analyzed the field data and helped write the manuscript. MA design the experiment, and coordinated and collected the field
data. TF helped with the experimental design and data analysis, and reviewed the manuscript. GHR helped with the experimental
design and reviewed the manuscript.

## Competing interests

The authors declare that they have no conflict of interest.

## Acknowledgements

The authors gratefully acknowledge funding from Canada Foundation for Innovation – John Evans Leadership Fund, Natural
Sciences and Engineering Research Council of Canada – Discovery Grant, and Agriculture and Agri-Food Canada – Agricultural
Greenhouse Gases Program 2, as well as assistance from Dick Puurveen and Sheilah Nolan.

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

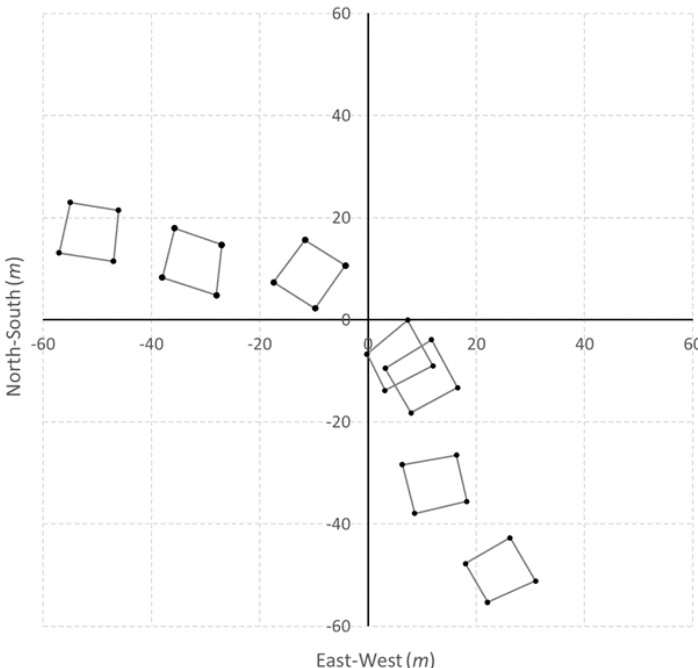

**Figure 1: Map of the synthetic source locations used in the study (polygons). The eddy covariance system was located at position (0,0).**

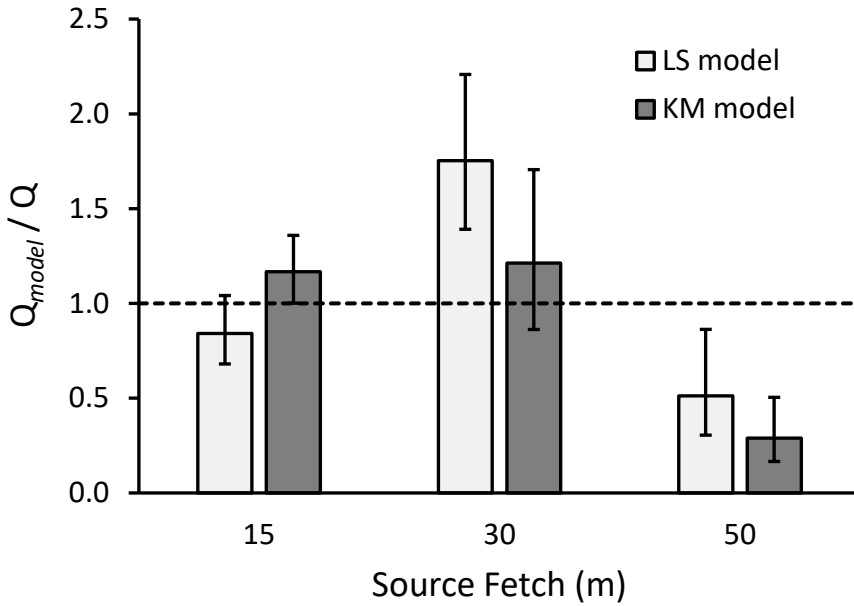


**Figure 2: Agreement ratio of the footprint model calculated emission rate ($Q_{model}$) to actual release rate ($Q$), grouped by source fetch of**
**15 m (n = 26), 30 m (n = 9), and 50 m (n = 24). Calculations are from the LS and KM models. The columns show the geometric mean,**
**and the error bars show the 95% confidence interval of the mean. The horizontal dashed line represents a $Q_{model}$ / $Q$ ratio of one, or a**
**perfect model calculation.**

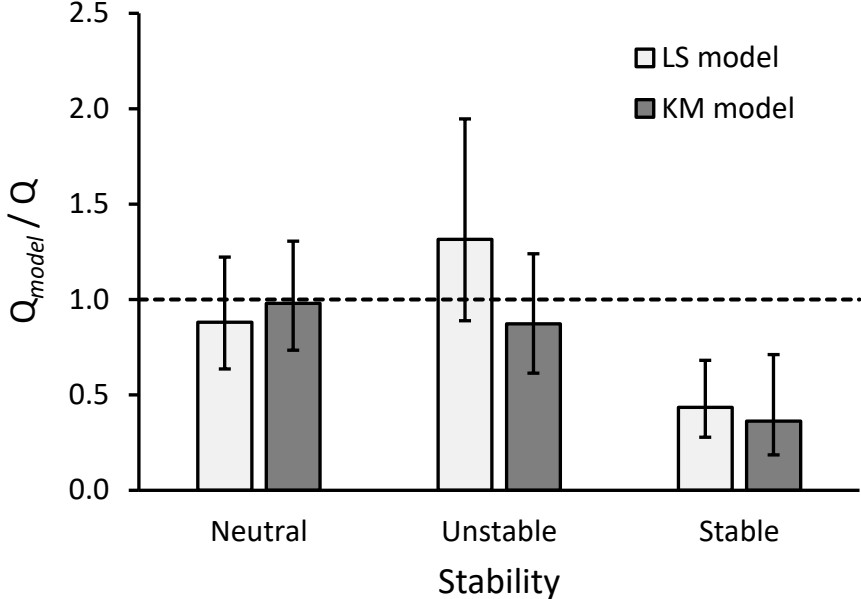


**Figure 3: Agreement ratio of the footprint model calculated emission rate ($Q_{model}$) to actual release rate ($Q$), grouped by atmospheric stability: neutral ($|L| > 60$ m), unstable ($0 > L > -60$), and stable ($60 > L > 0$). Calculations are from the LS and KM models. The columns show the geometric mean, and the error bars show the 95% confidence interval of the mean. The horizontal dashed line represents a $Q_{model}/Q$ ratio of one, or a perfect model calculation.**