# Peer review of "Field Testing Two Flux Footprint Models"

_Atmospheric Measurement Techniques, 2021_

## Author Response (AR1)

**Field Testing Two Flux Footprint Models**

**Trevor W. Coates et al.**

**Author's Response to Reviews**

**A) Author Comments Regarding Change in Analysis**

While addressing the reviewer's comments, we also made a change to the methodology of our study. The metric to evaluate the footprint models is the ratio of the model-calculated emission rate to the actual emission rate of the synthetic source ($Q_{KM}/Q$ or $Q_{LS}/Q$). In the original manuscript we calculated the arithmetic mean, the standard error of mean, and statistical significance assuming a normal distribution of the ratio data (i.e., traditional statistical inferences). It came to our attention that ratio (or normalized) data is more correctly evaluated using the geometric mean, with inferences based on log-transformed data (Fleming and Wallace, 1986; Limpert et al., 2001).

In the revised manuscript we re-evaluated our data using the geometric mean as the measure of central tendency, and calculated the 95% confidence intervals for the means using log-transformed data. Details are included in a new section, copied below.

**2.3 Statistical Analysis**

*The accuracies of the footprint calculations are evaluated from the ratio of the model calculated emission rate to the actual release rate: $Q_{KM}/Q$ and $Q_{LS}/Q$. With ratio data the geometric mean is a more meaningful measure of the central tendency than is the arithmetic mean (Fleming and Wallace, 1986), and we use the geometric mean to describe our ratio data. Confidence intervals for the geometric mean are calculated using the log-transformed ratio data, and then converted back to ratio units (Limpert et al., 2001). The confidence intervals (CI) are asymmetrical, and we report the upper and lower limits of the intervals.*

*Fleming, P.J., and Wallace, J.J.: How not to lie with statistics: the correct way to summarize benchmark results. Communications of the ACM. 29 (3): 218–221. doi:10.1145/5666.5673. S2CID 1047380, 1986.*

*Limpert, E., Stahel, W.A., and Abbt, M.: Log-normal distributions across the sciences: keys and clues. BioSci. 51, 341-352, 2001.*

This change did not alter the main conclusions or our study, nor did it alter the dataset included with this study. However, it did somewhat change some of the relationships between the two models, which is reflected in the modified discussion.

**B) Response to Reviewer Comments**

We very much appreciate the feedback, and thank the reviewers for their careful reading of the paper. We respond to their specific comments below.

**Reviewer 1 (RC1)**

RC1: 'Comment on amt-2021-106', Albrecht Neftel, 12 Jul 2021

RC1: This is a short paper and reports a comparison of two flux footprint models using an artificial $CO_2$ source of a limited areal extension as a known emission source. Technically I judge that everything is correctly made. The results show that both flux print models yield a recovery rates that is statistically not different from one.

I could sit back contentedly and rejoice that my simple Footprint Tool based on the KM algorithm still produces satisfactory results. Nevertheless, I think it is appropriate to address some warnings. The findings are based in total on 59 valid 10 minutes intervals. They were divided into three fetch-dependent groups. 10 minutes is a short time interval for EC analysis. Consequently, the variability in the recovery rate is large and the fact that the recovery rates are statistically not different from 1 cannot be a strong statement.

Author Response: There is little to disagree with in Dr. Neftel's statement.

Yes, 10 minutes (min) is a short interval compared to typical EC studies of surface fluxes, but it does fall within a broad 5 to 60 min averaging interval that has used in micrometeorological studies (e.g., footprint study of Kumari et al., 2020 used a 10 min interval). Given the short fetches in this study, and our experience of insensitivity in LS model accuracy with averaging intervals from 5 to 60 min, we think the conclusions of this paper are also valid for longer averaging periods. We did consider a 20 min averaging time, and found the longer interval resulted in a slight decline in the accuracy of our LS model calculations. But more importantly, the choice of a 20 min interval left fewer than half the number of good observation periods in our data, and accordingly larger statistical uncertainties.

We also agree that with our relatively small dataset, and large period-to-period variability in the accuracy of the footprint calculations, the finding that the KM and LS calculations are not different from each other is not a strong statement. We think this is noted in section 3 (e.g., *"This suggests that any systematic differences between the models in our study were obscured by the substantial period-to-period variability in the Q / Q calculations, and that the detection of model differences would require a much larger observational sample size than we were able to acquire."*). However, the conclusions are still of value. Prior to our experiment, we expected large differences between the KM and LS models, and the demonstrable value of the LS model predictions. It was a surprise not to see large differences, suggesting that experimentalists may not see the advantages of a more complex LS model given the period-to-period variability of real world data. This is useful information.

Our results suggest that much larger datasets would be needed to discriminate between footprint models for a configuration like ours. But we note the difficulty of generating large field datasets for this type of comparison. This study does not provide the final word on this subject, but hopefully our dataset (provided as a supplement to the paper) will provide a useful piece of that effort.

Kumari, S., Kambhammettu, B. V. N. P., & Niyogi, D. Sensitivity of analytical flux footprint models in diverse source-receptor configurations: A field experimental study. Journal of Geophysical Research: Biogeosciences, 125, https://agupubs.onlinelibrary.wiley.com/doi/epdf/10.1029/2020JG005694. 2020.

RC1: I had a look at the data given in the supplement. It is striking that three consecutive data points or one 30-minute value of the KM based recovery data for the 30m fetch group are clearly < 1 whereas the other four values are above 1, of course on average around 1. The bls based recovery rates for this group is clearly higher but does not reflect the distinction in two groups. I guess this is the typical behavior of real turbulence. This reminds me the flux simulation with a large eddy simulation approach that demonstrated the possibility of persistent structures lasting longer time that are inexistent in the KM or bls world. This information was presented during a workshop on ammonia measurements (Hensen et al, 2015). I recall the sentence*: From the LES simulations we can assume that for time averaging below 15 minutes integration the effect of streaky structures might be detectable on the plot scale. For multi hour averaging on the other hand, the effect might cancel out."*

Author Response: As Dr. Neftel notes, the period-to-period variability of our footprint calculations is large and difficult to relate to environmental variables. We agree with his simple explanation: "this is the behaviour of real turbulence". To be more specific, the KM and LS models are built on a representation of the atmosphere that is true (at best) in an ensemble average sense. They will not reflect the period-to-period fluctuations in our dataset. We have made this point in the final sentences of the manuscript, adding:

*"However, period-to-period variability is the nature of footprint calculations based on simplified models of atmospheric transport like the KM and LS formulations. These model calculations, which at best approximate*

*an ensemble average realization of the atmosphere, will not reflect the period-to-period fluctuations of actual measurement periods."*

RC1: Footprint corrections are always necessary in case a measured flux over areas with different emissions must be interpreted. The new generation of researchers are generally well trained in computing languages such as R I recommend the use of a bls model because it tends to force the user to think about the micrometeorological boundary conditions. A special package made by Christoph Häni is available

Author Response: We share Dr. Neftel's preference for the more sound LS modelling approach, and endorse the LS analysis package by Häni. However, our experience shows that increased processing time is a significant penalty with the LS calculations. In a field study associated with this project, we used an LS footprint model to calculate the EC flux contribution from a field surrounding the EC system. Analyzing two years of EC measurements took several weeks of processing. Some researchers may have difficulty justifying the processing time of LS models relative to the KM calculations given the model agreement seen in our data.

**Response to Reviewer Comments: Reviewer 2 (RC2)**

RC2: 'Comment on amt-2021-106', Thomas Foken, 01 Aug 2021

RC2: Footprint models are widely used, but there has been little validation of the models (Leclerc and Foken, 2014). Such publications are very rare, as experimental validation is very costly. This publication is the description of such an experiment. It compares an analytical model (Kormann and Meixner, 2001) and a Lagrangian model (Flesch, 1996;Flesch et al., 2004) – both well-known model concepts – with a tracer experiment. Although the paper is brief, no methodological shortcomings could be identified. It should be published in the present version, unless the following comments make it possible to add to it.

Author Response: We thank Dr. Foken for his encouragement. Yes, this type of experiment is difficult, and a great deal of effort and expense was required to get good (testable) observations (e.g., multiple source realignments in response to changing winds, cold weather freezing gas regulators, equipment issues). Regardless of these difficulties, in retrospect the weakness of the study is the limited number of observations. The most valuable aspect of this work is likely to be our dataset, which could be built upon by others and provide for a more robust examination of footprint models.

RC2: Thankfully, the measurement data were published in the supplement. A first look at the data showed me that with stable stratification the analytical model agrees particularly badly with the validation data, while the Lagrangian model delivers significantly better results. A similar result was found by Göckede et al. (2005) when comparing the models of Schmid (2002) and Rannik et al. (2004), also mentioned in the paper. Perhaps one could make an addition to the stability dependence analogous to Fig. 2 in Göckede et al. (2005) and thus enhance the contribution somewhat.

Author Response: We added Figure 3 to the manuscript showing the model comparisons versus stability. Dr. Foken is correct that the KM results are inaccurate in stable conditions (the mean $Q_{KM}/Q = 0.36$). However, the LS results are similarly inaccurate ($Q_{LS}/Q = 0.44$) in these conditions. It is interesting that these results disagree with the calculations made by Göckede et al. (2005), who found large differences in calculations from the two models in stable conditions. The similarity is also surprising given the substantial differences in the stable KM and LS footprint functions calculated by Wilson (2015).

We have added a reference to the Göckede et al. paper, indicating the surprising result that our measurements could not discriminate between the two models (end of the results section):

*"There are no clear patterns in terms of explaining the differences between the two footprint models based on environmental factors. Whether we separate the data by fetch or by stability, the results from the two models are not statistically different from each other. Windspeed, roughness length, and wind direction were also considered as factors to explain the model differences, but again, no pattern was observed. This lack of model differences was unexpected given the studies of Göckede et al. (2005) and Wilson (2015) showing large*

*differences in the calculations between analytical and LS models. This suggests that in our study, any systematic differences between the models were obscured by the substantial period-to-period variability in the Q / Q calculations, and that the detection of model differences would require a much larger observational sample size than we were able to acquire."*

RC2: At least in the further discussion of the data, attention should be paid to the positions of the maximum of the footprint. A possible explanation for the different agreement depending on fetch could be that the maxima of the footprint fit better with short fetch. This would be an investigation similar to that of Markkanen et al. (2009) for other model types and altitude ranges.

Author Response: The suggestion to look in detail at the footprint function (vs. fetch) is good, although we see this as of secondary importance given our simple objective of determining if the accuracy of the LS model was better than the KM model for our dataset (and to provide a short description of our dataset for others to use). Further analyses of the footprint functions would require substantially more analysis, and would duplicate the analysis of Wilson (2015). For example, Wilson's Figure 3 shows the difference between the KM and LS footprint functions for stable conditions in a configuration that has some similarity to ours ($z_0 = 0.01$ m, $z_{sonic} = 2$ m, $L = 25$ m). The peak of the LS footprint function is near $x = 20$ m, while the KM peak is near $x = 26$ m. Here the peak position is similar between the models, and midway between the short and medium fetches, however, it is unclear if this difference in peak position affected the $Q/Q$ ratios.

*Wilson, J.D.: Computing the Flux Footprint, Boundary-Layer Meteorol., 156, 1-14, https://doi.org/10.1007/s10546-015-0017-9, 2015.*

RC2: For the quality test of the eddy covariance data, no programme documentation should be cited, but either the original paper (Foken and Wichura, 1996) or the identical book publication (Foken et al., 2012).

We have made the suggested change regarding the citation for quality testing of the eddy covariance data.

---

## Author Response (AR2)

Dear authors,

the referees made only minor comments, which you have addressed in a satisfying way. However, I have identified an issue related to the changed statistical analysis in the revised version (see Comment 5). In addition there are some other minor issues that need to be addressed before the manuscript is ready for publication. They are listed below.

Best wishes
Christof Ammann
AMT Associate Editor

Dear Dr. Ammann,

Thank you for the constructive feedback on our paper. Below we list our responses to your comments.

EDITOR COMMENTS

1) Line 45: Was the mass flow controller specifically calibrated for CO2?

The mass flow controller (MFC, Aalborg GFC model) was newly purchased, configured for $CO_2$, and factory calibrated. As a gross check on the MFC accuracy, we monitored the weights of the $CO_2$ cylinders used for the gas release study (measured before and after use). Over the full-study, the total MFC based gas release total agreed with that calculated from the cylinder weight loss to within 5%.

We have added more information about the MFC on line 46:

> "… passed through a mass flow controller (GFC57 configured for $CO_2$, Aalborg Instruments and Controls, Inc. Orangeburg, NY, USA) …"

2) Line 47: What was the pressure in the manifold? Was it really much above ambient pressure?

The inlet pressure to the MFC was 150 kPa, and the maximum pressure drop across the MFC as given by the manufacturer is approximately 75 kPa for the flow rates used. Based on simple engineering calculations for a similar synthetic source, we assume the pressure loss across the manifold, hoses, and piping is small (as mentioned in Flesch et al. 2004). We thus estimate the manifold pressure as greater than 75 kPa (or 10 psid).

On a related point, we have modified the text on line 48-49:

> "We assumed equal flow rates from each outlet due to the high head loss across each outlet relative to the manifold pressure (following the argument made by Flesch et al., 2004)"

3) Line 65: What was the value range of the background fluxes? Are they negligible for the uncertainty of the results? Some of the measured EC fluxes for the large source distance were also quite small (according to the data in the supplemental material).

The background EC fluxes varied from -0.8 to +0.9 umol m$^{-2}$ s$^{-1}$.  During gas release periods the measured fluxes ranged from 0.3 to 118 umol m$^{-2}$ s$^{-1}$.  In most cases the measured flux during gas release was more than an order of magnitude larger than the background flux.  But as you mention, this was not always the case with the larger fetches (as seen in the supplemental file).  So yes, in these cases there will be a relatively larger level of measurement uncertainty in $Q_{LS}$ or $Q_{KM}$.  But these are also cases where the actual release rate is large, so that the uncertainty in the $Q_{LS}$/Q or $Q_{KM}$/Q ratios should still be relatively small.

4) Line 86: Unfortunately the link provided in Neftel et al. (2008) to access the ART footprint tool software is obsolete. You could add the following updated DOI link to access the tool: http://doi.org/10.5281/zenodo.816236

Thank you for this information.  Reference to the ART footprint software (Spirig et al. 2007) was added to the text on line 87, and the DOI link  to access the tool was included in the list of References.

5) Line 103-106: I agree that the geometric mean and corresponding asymmetric confidence interval are likely more accurate for the present data than the originally used artithmetic mean. However, the provided argument is not really appropriate. It is true that the ratio of two quantites with Gaussian error distributions shows an asymmetric error distribution (F-distribution). But in the present case, the error (uncertainty) of Q_KM/Q and Q_LS/Q are almost fully determined by the error of the nominator, because the denominator (Q) presumably has a negligible random error. This means that Q_KM and Q_LS themselves already must have an asymmetric error distribution, which is reasonable because they are calculated as a ratio with the footprint fraction (of the artificial source) in the nominator and the measured EC flux in the denominator. In addtion, the modelled footprint fraction may itself have an asymmetric error distribution because of the limitation to positive values and the special shape of the footprint function. Please reconsider and improve the reasoning/explanation for the use of the geometric mean. Apart from theoretical considerations, it could simply be argued that the values of Q_model/Q show a clearly asymmetric distribution and thus a logarithmic transformation is useful. It may additionally be useful to test whether the geometric mean and the median of Q_model/Q show similar results.

These are very good points, and indicate the potential for a more complex analysis.  And we seem to agree on the broad issue of how this data can be analyzed in a relatively simple way.  We want to make a few general comments:

- It is intuitive that our ratio data ($Q_{KM}$/Q, $Q_{LS}$/Q) are not normally distributed: the ratios are bounded at zero and unbounded at the top end.  It is intuitive that the arithmetic mean is not the ideal measure of central tendency.
- It is broadly accepted in statistical texts that for normalized data (like our ratios), the geometric mean is a better measure of central tendency than the arithmetic mean.

- The big-picture conclusions of our study (e.g., the large period-to-period variability in model accuracy, the lack of statistical differences between the LS and KM models, the poor performance of both models at large fetches) hold true regardless of whether we analyze the data in terms of arithmetic or geometric means (and associated uncertainties).  This suggests insensitivity in our conclusions to the exact details of the statistical analysis.

With the goal of concisely expressing a simple, but scientifically reasonable statistical approach, we have taken your suggestion and simply said that our ratios are asymmetrically distributed, and have used a logarithmic transformation of our data (line 102):

> ### 2.3 Statistical Analysis
>
> *The accuracies of the footprint calculations are evaluated from the ratio of the model calculated emission rate to the actual release rate: $Q_{KM}/Q$ and $Q_{LS}/Q$.  These ratio data are asymmetrically distributed, and a logarithmic transform of the ratios is used when making our statistical comparisons.  Thus, the geometric means of the emission ratios is our measure of central tendency.  Confidence intervals for the geometric mean are calculated using the log-transformed ratio data, and then converted back to ratio units (Limpert et al., 2001). The confidence intervals (CI) are asymmetrical, and we report the upper and lower limits of the intervals.*

6) In the Introduction or Discussion sections, you may want to include the following references to other studies using an artificial source to test footprint models:

Heidbach et al.: Experimental evaluation of flux footprint models. Agricultural and Forest Meteorology, 246, 142-153, 2017.
Kumari et al.: Sensitivity of Analytical Flux Footprint Models in Diverse Source-Receptor Configurations: A Field Experimental Study. JGR Biogeosciences
125(8),e2020JG00569, 2020.

This is a good suggestion.  It shows that our experimental design follows a well-accepted approach.  We have added the Heidbach et al. reference as suggested, and have also added a reference to a field study conducted by some of our team:

> *"This field study compares the accuracy of the KM footprint model with a more rigorous LS model.  The motivation for this study was the question of whether the accuracy of the LS model was sufficiently better than the KM model so as to justify a more complex LS application.  In this experiment we released gas at a known rate from a small synthetic area source and measured the vertical gas flux at a downwind location using the eddy covariance technique.  The KM and LS models were then used to calculate the source emission rate from the measured atmospheric flux.  The accuracy of those calculations is examined in this report. This follows the approach of Heidbach et al. (2017) and Coates et al. (2017) in their experimental evaluation of footprint models."*

With the addition of the Heidbach et al. reference, we took the opportunity to acknowledge one of the results demonstrated by Heidbach et al. in our results section (line 122):

> *"Based on the calculations of Wilson (2015) and Heidbach et al. (2017), we had hypothesized that there would be substantial differences between the two models at the shorter fetch, with the LS model being more accurate than KM due to a better representation of horizontal turbulent transport, which is particularly important for defining the footprint at short fetches. However, this is not the case in this study."*

7) Figure 2: Indicate in the figure caption the number of data (n) in the three classes.

The caption for Fig. 2 now includes information about the number of data in each of the three classes, as noted below (lines 233-236).

> *"Figure 2: Agreement ratio of the footprint model calculated emission rate ($Q_{model}$) to actual release rate (Q), grouped by source fetch of 15 m (n = 26), 30 m (n = 9), and 50 m (n = 24). Calculations are from the LS and KM models. The columns show the geometric mean, and the error bars show the 95% confidence interval of the mean. The horizontal dashed line represents a $Q_{model}$ / Q ratio of one, or a perfect model calculation."*

8) Add a (short) caption to the Table in the supplementary material

The following title has been added to the supplementary material.

> *"Supplementary Table. Final data set used for footprint analyses following application of quality control criteria."*

---

## Author Response (AR3)

Dear authors,

You have addressed most of my comments in satisfactory way. However, I have two minor remaining points that should be addressed before submitting the final paper version. They are listed in the following (referring to numbering in the author response).

Best regards
Christof Ammann
AMT Associate Editor

Dear Dr. Ammann,

Thank you again for the constructive feedback on our paper. Below we list our responses to your comments.

2) line 48. I assume that the pressure values in your response refer to "gauge pressure" (pressure above ambient pressure) and not absolute pressure, which would be standard for SI units. Anyway it would be useful to indicate (estimated) values for the manifold pressure and/or the pressure drop across the outlets in the paper.

We added the following text, giving an estimate of the pressure loss across each outlet.

> *"We assumed equal flow rates from each outlet, which requires the gas outlets be identical and the pressure loss across each outlet to be much greater than the pressure loss along the source piping (Flesch et al., 2004). We estimated pressure losses using simplified equations for pipe flow, assuming incompressibility and a re-entrant type outlet shape (Fox and McDonald, 1985). For our most commonly used release rate of 90 L min$^{-1}$, the pressure loss across the outlets is approximately 5,000 Pa whereas the loss along a 10 m pipe section is only approximately 40 Pa."*

These calculations are described in the cited reference (Fox and McDonald, 1985), in their chapter on internal incompressible viscous flows. We did not add the estimated pressure on the outlet side of the mass flow controller (MFC), as we do not know with any certainty the pressure drop across the MFC.

3) I do not agree with your argument that the uncertainty of Q.model/Q ratios are small if the actual release rate Q is large. The (relative) uncertainty of Q.model/Q is largely determined by the uncertainty of Q.model. And the uncertainty of Q.model is limited by the uncertainty of the flux difference (F.measured - F.background) independent of Q. So in cases when F.measured and F.background are of similar size, the uncertainty of the background estimation can play a significant role. This should be mentioned in the manuscript.

We agree that the fractional uncertainty in $Q_{model}$ equals the fractional uncertainty in $F_{meas} - F_{back}$ (ignoring uncertainty in the footprint model calculations). And we agree that the fractional uncertainty in $F_{meas} - F_{back}$ will increase as $F_{meas}$ falls to levels near $F_{back}$. We do appreciate that for some of our 50 m fetch emission calculations, this uncertainty can become large. We have added the following text:

> *"At the 50 m fetch the measured EC fluxes were smaller than was measured at the shorter fetches, and in some cases the measured flux fell to a level near the background landscape flux*

*(e.g., five periods had a measured flux that was less than five times the magnitude of the background flux). This was despite maximizing the gas release rate for the larger fetches. The result is that for the larger fetches there is increased measurement uncertainty (relative) in the flux signal from the gas release, and increased uncertainty in $Q_{KM}$ and $Q_{LS}$. Some of the relative uncertainty we see in $Q / Q$ for the 50 m fetch is likely due to this factor."*